# Promoting Evidence Based Nutrition Education Across the World in a Competitive Space: Delivering a Massive Open Online Course

**DOI:** 10.3390/nu12020344

**Published:** 2020-01-28

**Authors:** Simone Gibson, Melissa Adamski, Michelle Blumfield, Janeane Dart, Chiara Murgia, Evelyn Volders, Helen Truby

**Affiliations:** Department of Nutrition, Dietetics and Food, Monash University, Notting Hill, VIC 3168, Australia; Melissa.adamski@monash.edu (M.A.); michelle.blumfield@monash.edu (M.B.); janeane.dart@monash.edu (J.D.); Chiara.murgia@monash.edu (C.M.); evelyn.volders@monash.edu (E.V.); helen.truby@monash.edu (H.T.)

**Keywords:** distance education, global education, health promotion, internet, nutrition misinformation, online learning, social media

## Abstract

The internet is the fastest growing source of nutrition information for consumers. Massive Open Online Courses (MOOCs) provide and avenue for nutrition professionals’ urgent need to respond to consumer demand for low-cost, accessible and engaging information. This research aimed to evaluate learner participation and perceptions in an evidence-based nutrition MOOC and provide recommendations for engaging international online lay audiences. Learners completed pre and post course surveys including quantitative and open-ended questions. Pre-course surveys collected demographic data, prior nutrition knowledge and motivations for doing the course. Post-course surveys evaluated their preferred learning modes and learners’ opinions of the course. Quantitative were analyzed using descriptive statistics. Conventional content analysis was conducted on learners’ responses to open-ended survey questions using an inductive approach. Learners represented 158 countries from a range of educational backgrounds. There were 3799 qualitative comments related to learners’ learning and course content preferences. Qualitative analysis identified key themes related to (1) online interaction, the (2) value of the evidence presented by nutrition experts and (3) the course structure and practical aspects. Divergent opinions were expressed within these themes. Satisfying the needs of large international audiences with diverse backgrounds is challenging in promoting sound evidence-based nutrition messages. MOOCs provide a means for delivering evidence based global nutrition education in the online space crowded with food advertising and nutrition conjecture. Recommendations are made as to how to construct and engage diverse on-line audiences.

## 1. Introduction

There has been an explosion of easily accessible health-related information over the past twenty years with the public now able to access a busy fast-moving stream of information via the internet. Near universal access to the internet has resulted in rapid transmission of nutrition information from both regulated and unregulated sources who compete for consumers’ attention. It is not surprising that with the plethora of nutrition information available, consumers are confused [1,2], while doctors feel ill-equipped to meet consumer need [3]. Consumers can be influenced by on-line nutrition messages promoted by celebrities and exponents of pop culture [4] which can adversely affect health literacy and behaviors [5,6]. Indeed, some diet followers have come to regard their nutritional beliefs with religious fervor [7].

Complicating the communication of nutrition science and public health messaging are the often-contradictory range of messages communicated by researchers. Reductionist approaches focusing on research isolating single nutrients from whole diets in order to elicit causation, can lead to dissemination of oversimplified statements regarding healthy and unhealthy foods [8]. Nutrition guidelines were designed to provide consumers with evidence-based information to improve eating behaviors [9], however people are tired of repetitive government-initiated nutrition messages [10] and there is an appetite for consumers to have absolute recommendations [8]. In an era of “alternative facts”, nutrition science is particularly vulnerable to misinterpretation [11]. The lay public prefers clear and definitive guidance, rather than messages of moderation, variety and balance.

Seventy-two percent of Americans use the internet to search for health information [12] and this is the fastest growing source of health information [13]. This instant, free and convenient source has significant advantages over traditional avenues of health information such as seeking advice from qualified health professionals such as making appointments, waiting, travelling to and paying for a service.

In order to provide accurate nutrition information and assist in addressing population health concerns such as rising obesity rates [14], health professionals need to have greater reach and be able to communicate their messages clearly [15]. Adopting a delivery format that the public want and are willing to engage with—the internet—provides a range of untapped opportunities for education. Online evidence-based nutrition courses developed by experts helps to do this [16] and are proven effective in dispelling myths as well as understanding sub-group needs [17] who regard health professionals as ”trusted sources”. Nutrition professionals need to respond quickly to consumers’ demand for low-cost, accessible and accurate information or risk being overtaken by non-experts readily provided advice on-line [18]. Massive Open Online Courses (MOOCs) are an initiative that can contribute to filling this void.

MOOCs challenge traditional fee-paying education systems, reaching both the general public and being used in professional training programs. They can potentially overcome many barriers for health education [19] including no cost to learners, no requirement for pre-requisite skills and knowledge, plus the advantage of learners becoming part of an international community and having access to academics at university, perhaps for the first time. The World Health Organisation Decadal Action on Nutrition suggests they could be a vehicle to overcome many barriers for nutrition education [20].

There are over 11,000 MOOCs offered by more than 900 universities around the world (EdSurge) using platforms including Coursera, edX and FutureLearn. There are courses in every discipline offered by universities including business, education, art and design, engineering, science, humanities and health and medicine. MOOCs offer learners a range of levels of certification, from short courses with optional certificates of completion, to microcredentials and whole degrees.

Online learning, whether it be via MOOCs, social media or accessing websites, differs from traditional instructivist models. MOOCs typically draw upon connectivist learning theory where learners choose what they learn and construct their own meanings of learning from a range of sources which they connect in their own way [21]. This focus on knowledge creation, rather than remembering and reciting facts, is consistent with lifelong learning principles and is suitable for the wide range of global MOOC learners [22]. The principles of connectivism that relate to MOOCs learning include learners contributing and accessing diverse opinions, learners being motivated to learn more than they currently know, learning activities aiming to be current and accurate and learners are able to make their own decisions and connections [23]. A responsibility for MOOC deliverers is to foster connections to facilitate this learning [24]. 

Learners enrolled in MOOCs come from a range of backgrounds from around the world, and although they typically have a university qualification, and are over 25, their demographics can vary in terms of nationality and country of origin, education level with a wide range of motivating factors for doing the MOOC [25]. The topic of the MOOC is likely to influence these factors also.

Understanding learners’ patterns of interaction and engagement is required to ensure MOOCs meet learners’ needs [26]. Much research to date has focused on data analytics, but to be truly learner-centric, educators need to discover what learners think, with learner satisfaction an essential component to ensure MOOCs continue to reach large online audiences [27]. 

There is a gap in the literature for nutrition professionals delivering nutrition MOOCs and other online nutrition education programs, about learners’ perceptions and how to engage on-line learners in this competitive space. The MOOC in this study, named “Food as Medicine”, was ranked in the top 100 MOOCs worldwide in 2019 so investigating its learners’ perceptions is warranted. The aim of this paper is to (1) provide an overview of a nutrition MOOC and its’ content; (2) describe course participation rates and participant backgrounds; (3) report on learners preferred learning modes and content, and (4) share our learnings and recommendations.

## 2. Materials and Methods 

An evidence-based nutrition-specific MOOC designed for the general public was developed for the United Kingdom (UK) FutureLearn™ platform. Entitled “Food as Medicine” it was developed by a steering group involving health professionals and academics from dietetics, nutrition science, medicine and education backgrounds. The title was designed to be attractive to the public, with #foodasmedicine and #foodismedicine having growing acceptance on social media and used frequently by key on-line influencers. The aim was to make the content broadly applicable to an international lay audience while addressing key health priorities and current nutrition topics in the media. The 3-week course emphasis was on whole foods and dietary patterns rather than individual nutrients, aiming to promote health literacy and eating practices. No prior knowledge of nutrition, health or science was assumed.

The backbone of the course design was consistent with the First Principles of Instruction, encouraging problem-centered and applied learning, and adult learning theory [28]. Based on the connectivism principles [24], discussion forums were used for learners to interact and connect. Learners were encouraged to participate in online conversations and were prompted with questions to stimulate discussion and apply the content to their own experiences. Stimulating learner engagement and relating to their experience was a priority in the MOOC design. Throughout the course learners were frequently asked to contribute their thoughts and experiences related to the topic and provide comments in response to each others’ input using discussion forums. They were encouraged to consider how they would implement their learning and describe challenges and enablers they face, and share this with others in the course. 

Content was organized into a week by week format (Appendix A) covering a range of issues in nutrition and health. Week one focused on the role of evidence in nutrition, the food matrix and topical issues such as superfoods and inflammation. The second week discussed foods and different body systems, while the final week covered nutrition controversies, implementing food choices using dietary guidelines and portion sizes, and tips on how to evaluate nutrition information. Learners were expected to spend approximately four hours per week doing the course. The course consisted of a range of teaching and learning strategies including articles, videos, animations and infographics, surveys, activities and discussion points. Quizzes were interspersed throughout the three weeks so learners could test their own knowledge and learning. Certificates of participation were available to those who completed all modules of the course for purchase to suit those with more academic aspirations [26].

The online MOOC commenced in May 2016 and was ‘live’ for 3 weeks, however learners who enrolled by the finish date had access to the course content indefinitely. Ethics approval was obtained by the Monash University Human Research and Ethics CF16/905–2016000470 and learner details were anonymous.

The research team was involved in the developing course content and/or were presenters throughout the MOOC. The research team were experienced accredited practicing dietitians and/or registered nutritionists with educational/pedagogical and/or research expertise.

### 2.1. Data Collection

Learners completed an optional survey at the commencement of the course collecting basic demographic data including age, nationality, gender, previous online course participation, reasons for doing the course and a broad description of prior professional knowledge of the topic. An optional post course survey was available at the final step in the course to elicit overall learning and satisfaction. Questions included preferred learning modes using 5-point Likert scales and open-ended questions asking learners opinions on their least and favorite parts of the course, which were part of the standard FutureLearn evaluation for all their MOOCs. Questions were not mandatory. These data were collected by 12 July 2016, within 10 weeks after course commencement.

### 2.2. Data Analysis

Descriptive statistics were used to analyze participation rates, learner characteristics and learning mode preferences and reasons for taking the course. Post course evaluations were qualitatively analyzed using conventional content analysis [29] for responses to the open-ended questions “What was your favourite part of the course, and why?” and “What was your least favourite part of the course, and why?” to elicit an overview of learners’ experiences of the course. Text responses to each of these questions were independently coded by two researchers respectively (EV, JD, CM and MA) who came together to reduce codes to categories. All researchers then collaborated with a fifth researcher (SG) to refine categories and then develop themes. To help minimize bias, researchers who analyzed “least liked” responses were not involved as course presenters, and comments eliciting feelings of frustration and disappointment were discussed openly and purposefully included in the analysis. Appendix A contains the Qualitative Research Review (RATS) Checklist.

## 3. Results

### 3.1. Pre-Course Evaluation: Learners’ Backgrounds and Course Engagement Metrics

There were 62,144 people who registered for the MOOC, from 158 countries, with 41% from Australia, 22% UK, 7% USA, 2% Canada, 2% New Zealand and 27% from other countries representing Europe, Africa, Asia and South America. Twenty-two percent were aged 18–35 and 40% were aged over 56 years. Of the 12,500 learners completing the optional pre-course survey, the majority (54%) had not taken an online course previously. Previous experience in nutrition ranged from none (60%) to some learners having studied the subject at university (11%) or working in a nutrition-related field (18%). Reasons for enrolling in the MOOC as identified by strongly or slightly agree responses were: wishing to learn about the subject (*n* = 12,010, 99%), and the flexible MOOC format suiting their lifestyle (*n* = 9309, 80%). Approximately half of the learners (*n* = 6126, 54%) wanted to improve their job performance or career prospects.

Of the total number of learners who joined, 25,605 (41%) were classified as ‘active learners’ as defined by some type of engagement such as ‘likes’, with 8312 (32%) of this group responding to at least 50% of the learning steps. Overall 8978 learners (13%) fully completed the course, as defined by some on-line activity at the end of the course. Throughout the online discussion forums, there were approximately 105,000 comments made by 9100 learners, with 162,000 “likes” by learners of their fellow learners’ comments.

### 3.2. Post-Course Evaluation: Learning Preferences

The post course evaluation survey was completed by 2341 learners (28% of course completers as defined by on-line activity in week 3). Most learners enjoyed reading articles (92%), watching videos (93%), following links to related content (88%), doing quizzes (85%) and reading comments by other learners (73%). About half (54%) of learners said they liked discussing topics online with other learners. “Disliked” items accounted for less than 2% of all responses, except for reading comments by other learners (5%) and discussing things online with other learners (8%).

There were 3799 qualitative responses, ranging from single word answers to responses with several paragraphs and provided rich data for analysis. Qualitative analysis of the data resulted in the development of seven themes related to learners’ perceptions of the course: (i) structure and course design, (ii) learning by video, (iii) the role of the expert presenter, (iv) evidence-based underpinning of information, (v) learners’ online interactions, (vi) interest in content topics, and (vii) impact on knowledge or behavior. Within each theme were divergent views.

#### 3.2.1. Structure and Course Design

The variety of learning and teaching methods and tools utilized in the MOOC were positively received. These included articles, quizzes, polls, videos, debates, demonstrations, weekly feedback videos and activities that learners were encouraged to relate to their own diet and eating habits. Learners reported that expectations and explanations were clear and they valued the flexible, self-paced approach. The opportunity to ask questions and interact was identified as a strength of the course design.

“Getting us to ask questions and reflect, that’s when realisations happen rather than following things blindly or not truly knowing what/whom to believe”. (Learner, response to the ‘Most Liked’ comment).

“Small chunks, so easy to do a little at a time and fit in with a busy life” (Learner, response to the ‘Most Liked’ comment).

#### 3.2.2. Learning by Video

Videos were the most popular delivery format, consistent with the quantitative survey results. Learners stated the content was engaging, easy to absorb and they enjoyed watching the variety of presenters and demonstrations. Keeping the videos brief (<10 minutes) supported learners to remain interested. Negative comments related to difficulties with internet access and speed, annoying introductory music and computer-generated subtitles occasionally out of sync with the visual content. 

“Animations and videos, because I can understand the information more easily” (Learner, response to the ‘Most Liked’ comment).

#### 3.2.3. The Role of the Expert Presenter

Many learners appreciated the opportunity to learn from highly qualified experts in their fields from a reputable university. Learners valued the personal traits of the presenters such as enthusiasm and friendliness, balanced with credibility and authority. Weekly feedback videos from the course leader and mentor were described as fun, light-hearted and clarified some of the issues raised during the week.

“I just found the information about how to get credible sources to be very relevant... There is misinformation overload out there, and it’s sometimes hard to know how to get good information. It was also useful to know how to vet a credible dietitian or nutritionist.” (Learner, response to the ‘Most Liked’ comment)

However, some learners distrusted the experts. Defining an “expert” was controversial learners who did not view the presenting nutrition scientists with PhDs and/or accredited practicing dietitians as being authorities in the field of nutrition. These learners believed this to be a narrow focus, viewing themselves as also being experts and undervalued for their input. A small number of learners would have preferred the course to be presented by naturopaths and alternative medical practitioners.

“The strong emphasis that only professionals should be listened to—like we aren’t so intelligent.” (Learner, response to the ‘Least Liked’ comment).

“It was a complete let-down. Out of date information, especially coming from the dietician” (Learner, response to the ‘Least Liked’ comment).

#### 3.2.4. Evidence Based Underpinning of Information

The theme of evidence-based information attracted discourse for the most and least liked categories. Many learners appreciated the recommendations, which were regarded as sensible and supported by high quality evidence. They found the MOOC clarified their understanding of controversial nutrition information. However, some learners viewed the information as outdated and merely following government guidelines, which they regarded as unreliable. Many learners reported their fellow learners lacked insight into the value of scientific evidence, which was a cause of frustration in their prolific comment posting.

Learners appreciated the questioning approach adopted by the course presenters who encouraged learners to question and reflect on nutrition information while providing advice on how to evaluate scientific evidence.

“I particularly liked the balanced view and the fact it has equipped people with life skills in evaluating information and sources, enabling informed decisions.” (Learner, response to the ‘Most Liked’ comment).

#### 3.2.5. Learners’ Online Interactions

Some learners were very engaged in the on-line discussions and described them as being a valuable addition to the course content. They enjoyed the global nature of the learning community and the information posted by peers.

“I liked the fact that I was co-learning with people from all around the world. I found many gems of wisdom, different perspectives and useful extra information from other learners”

“Some comments by participants demonstrated their level of expertise in the area which added to my experience… thank you to all” (Learner, response to the ‘Most Liked’ comment).

Contrastingly, others regarded some of their fellow learners’ posts as rude, negative, opinionated, self-validating and unrelated to the course topic. The large volume of comments also discouraged some learners from participating in online discussions and thus some did not engage with other learners at all. Others reported only just reading the first two or three posts before moving on, or only followed the course mentor’s posts. Some learners were uncomfortable posting comments due to inexperience, shyness and fear of responses from their fellow learners.

“Some of the student comments were vapid, self-centered and added little to course knowledge, but I supposed helped their self validation” (Learner, response to the ‘Least Liked’ comment).

“Being expected to write comments, which I didn’t. I just hate writing anything that can be viewed by anybody and everybody. I don’t want to share my opinions, and I’m not interested in other people’s.” (Learner, response to the ‘Least Liked’ comment).

“When people who haven’t studied nutrition etc. comment, is it hard to tell whether it’s solid information or where they got their information from. Yet the course kept encouraging those comments and they confused me” (Learner, response to the ‘Least Liked’ comment).

#### 3.2.6. Interest in Content Topics

Learners found the topics to be diverse and relevant to their needs, with the exception of nutrition and pregnancy, which mostly males and older people found less interesting. Learners appreciated the inclusion of what was viewed as ‘cutting edge’ information such relating to inflammation, gut health, the microbiome and nutritional genomics. 

“Genomics made me realized how we are all very different and that one size does not fit all” (Learner, response to the ‘Most Liked’ comment)

Overall feedback regarding the MOOC content from the least liked category ranged from it being too basic, too complicated, too scientific and too nutrition related. Some learners felt the course title ‘Food as Medicine’ did not reflect the course content in that they were not expecting information regarding nutrition, dietary patterns and chronic disease prevention.

#### 3.2.7. Impact on Knowledge/Behavior

Many learners reported that their knowledge improved, and they were able to personalize the MOOC and utilize practical recommendations to apply to their eating behavior. This included overall changes in eating habits for themselves and their families, increased diversity of foods consumed and modifications to portion sizes. Some learners stated they were considering future study in nutrition as a result of doing the course. Others stated they learnt nothing new and the information was too basic.

### 3.3. Outcomes

From a synthesis of the qualitative and quantitative analyses, our top tips for creating an engaging nutrition MOOC are in Table 1. These concepts could be applied to other MOOC subjects targeted to a large international audience of the general public where a globalized consideration during content development is essential. Connectivist learning theory helps to explain the diversity of learning experiences, including why only some learners value expert opinions [21]. Acknowledging online learner behavior is complex and influenced by other learners’ input and opinions, which learners make connections with other information sources outside of the course and that online relationships facilitate learning [22] is imperative.

## 4. Discussion

Designing education for such a large number and great diversity of lay learners was immensely challenging. Learners predominantly enrolled because they were interested in the topic, consistent with other MOOCs [30]. Our learners’ reported diversity in their previous experience of nutrition-related education, which created large differences in their satisfaction with the depth of content. For nutrition professionals, as nearly everyone has some experience of preparing food and eating, as well as all MOOC learners having access to the internet, managing diverse and sometimes strongly held opinions required diplomacy as well as science. This diversity is reflected in the MOOC literature, with MOOC learners in a single course representing different ages, genders and nationalities as well as having differing motivations and goals for pursuing study and varied learning preferences [26].

Educational design is often compromised in MOOCs which frequently fail to be consistent with instructional design principles including problem-centered learning, drawing on learners’ previous experience and knowledge, provision of opportunities for learners to apply new knowledge, promoting collective and collaborative learning, supporting learners with different needs, providing authentic resources and expert feedback [31]. The development of this MOOC involved careful planning and development, and was approached using robust pedagogical and educational underpinnings to the course structure and design.

Communicating with mass audiences across different countries, time zones, experience and learning backgrounds posed the greatest challenge. Providing extension reading material and video links aimed to cater for those seeking deeper understanding of topics, while the real-time weekly feedback videos enabled controversial and pertinent online discussion topics to be addressed. Weekly feedback videos provided a more personalized touch, which is difficult to achieve in massive online education environments where attention to individual learner input is impossible.

The confidential post-course evaluation provided an avenue for otherwise silent participants who did not participate in course discussions to contribute feedback. These data provided a richness and depth in our understanding of learner engagement. It can be tempting to base opinions on the most vocal learners who posted prolifically throughout the course and this research identifies that online discussions did not always represent the majority of learners’ views. Challenges for learners regarding online posting included many learners’ reluctance to contribute their thoughts and experiences in a public forum due to their perceptions that their comments may provoke criticism from other learners, and the overwhelming nature of this environment, consistent with other MOOCs evaluations [30].

Discussion forums are essential for connecting learnings and are a key principle in connectivist MOOCs, however involving such large numbers of learners pose significant challenges for MOOC deliverers. Although the concept of MOOCs encourages learner interaction, this was frustrating for many learners seeking only evidence-based information from a university.

MOOC learners have been recognized as having different preferences for engaging in MOOCs and have been classified as active participants, passive participants and lurkers [23]. As these descriptors imply, active participants contribute to discussion forums and connect with other learners. Passive participants dislike contributing to discussions and are less likely to see the value of fellow learners’ contributions. Lurkers on the other hand may be active in terms of learning throughout the course but choose to remain invisible to other learners [23]. Consistent with our findings, factors such as confidence and motivation play a role in determining how learners engage in MOOCs [23]. Active participants are more likely to have undertaken MOOCs previously and are important players in generating discussion and role modelling behavior consistent with connectivism principles [23].

Promoting discussions also needs to be balanced with the risk of further spreading misinformation. Monitoring such a large volume of comments required significant resources and previous studies have reported that discussion forums lack focus [30]. Further research into the value of online discussions, including both learner satisfaction and learning outcomes would assist in developing effective strategies for interactive MOOC design. Technical improvements in tools to monitor engagement are required rather than just ‘clicks’ and ‘likes’ may better represent actual learner engagement.

Most learners who completed the course survey highly valued the expertise of the presenters and evidence-based principles promoted by the course. Videos were popular, which is consistent with research indicating that short videos (running less than three minutes) that include showing instructors’ faces who speak with pace and enthusiasm enhance engagement [32]. However, consistent with connectivism theory [22], others preferred the advice of their fellow learners to that of the presenters, indicating that professional credentials alone do not automatically lead to credibility by consumers. Distrust of nutrition professionals has been documented for decades [10,33]. Studies indicate that repetition, rather than expertise or credibility, has a greater influence of perceptions of “truth” [34]. The art of communicating diet and nutrition science remains challenged in how to engage in meaningful debate with the public and compete for a respected voice in the online space.

The negative comments regarding the evidence-based information presented in the MOOC are reflective of trends in society’s distrust of nutrition science and government guidelines [2,11], and also of other scientific issues, such as climate change and vaccination effectiveness [35]. Maintaining learners’ trust is crucial and many learners valued the credibility of the MOOC due to the reputation of the University. However, presenting evidence-based information needs to be delivered with diplomacy as consumers dislike being told what to eat [2,10], and it appears this included MOOC learners. This invites the question, were some learners seeking to learn new information, or did they view the MOOC as an opportunity to share their own beliefs? Alternatively, perhaps the style in which the information was presented failed to resonate with specific groups as demonstrated by comments relating to the perceived lack of respect for their own knowledge and forum contributions. Understanding online learner behavior is essential for successful communication of evidence-based messages. Utilizing theoretical frameworks such as intentional social action provides explanations for some of our findings whereby influences such as subjective and group norms, as well as social identity, motivate users to contribute online [36].

Evaluating the impact on learners’ knowledge and behavior is difficult to measure which is a criticism of MOOCs as an educational medium [31,37]. Many learners reported their knowledge and dietary practices improved as a result of doing the course, including eating a wider variety of foods and more fruit and vegetables. However, these outcomes were not evaluated.

Some topics were unexpectedly popular, such as portion sizes and the food matrix. A large number of learners stated these topics and video demonstrations enabled them to translate the information to everyday life. Nutrition educators need to remain mindful that basic information, if clearly and engagingly presented, is still of value and can fit within complex material. This may have resonance considering vegetable consumption in Australia where less than 10% of the population achieve recommended vegetable serves [38]. Providing information on more contemporary topics such as nutrition and inflammation also satisfied learner interest [39].

Consideration should be applied when translating results of this research to other MOOCs. Learners were predominantly female and older than most MOOCs learners, which was supported by the high number who stated they were retired and now had time to investigate topics they want to learn. This may have influenced lower ratings on particular nutrition topics and preference for not posting on forums. However, it does indicate that age is not a barrier for online learning and females aged 50–79 comprise over 30% of the Australian population [40], and have a great capacity to influence dietary and eating habits within families and across generations.

The strengths of this research include the large international sample size and the breadth of responses to evaluation questions. However, summarizing the content of such a large text-based dataset provides only an overview of learners’ perceptions. Limitations with data collection are inherent to MOOCs and include: post-course evaluation data did not capture the perceptions of learners who left the course early. These early non-completers may be more likely to provide negative comments nor do we know the reasons them for not continuing. Not all learners completed the post-course survey and due to the confidential nature of data, it was not possible to discover reasons for non-participation. Engagement metrics using ‘clicks’ were used to measure completion on the final unit of study, and using this metric 13% of learners completed all modules. This is comparable to other MOOCs (FutureLearn data, 2016), where attrition is regarded as inevitable but there is no doubt a need for a better method for measuring completion.

## 5. Conclusions

Meeting the needs of massive global audiences is challenging, but understanding learner needs is imperative to deliver content that enhances learning. There is benefit and value in using a MOOC as a means of dissemination of evidence based general nutrition information. Consideration for connectivist learning theory embraces learners’ knowledge, experience and propensity to participate in online discussions is key to ensuring learners feeling included in the online learning environment. We recommend evaluating learner feedback throughout the course so end of week feedback videos can promptly address learner queries and concerns and address misinformation. Furthermore, adapting the course for subsequent iterations can improve future learner experiences and the quality of discussion input.

## Figures and Tables

**Table 1 nutrients-12-00344-t001:** Top tips for successful Massive Open Online Courses (MOOCs).

Course Elements	Strategy
**Structure, content and course design** Ensure currency and relevance to target audience	Be clear about the message you are planning to deliver: what is the ‘course philosophy’ Consider Life-stages of target group, their technological abilities and learning styles Consider professional and educational backgrounds Acknowledge different motivations for doing course Incorporate quizzes with answers to support learning Scaffold learning
**Learning by video** Professional and high-quality delivery	Music, graphics, animations, video and photo quality all contribute to a polished and well-received product Ensure subtitles are available but do not interfere with video content Use real time video feedback to engage with and clarify discussion topics Video duration <7 minutes in length
**Role of the expert presenter** Engaging, approachable, knowledgeable and non-judgmental	Considerations: Wear bright single tone colors Smile Speak in conversational tone—do not read off a script Acknowledge a variety of points of view
**Evidence based underpinning of information** Ensure scientific content is catchy and engaging, topical and current	Provide sufficient background information so learners feel able to proceed Acknowledge international differences by using guidelines from a range of countries by linking to international organizations alongside anecdotes Acknowledge differences in terminology and be generic with brand names Include visual demonstrations Ask learners to reflect on own situation and current practices Encourage learners to apply learning to their own contexts Further in depth information can be provided in additional links and downloads for those who want additional learning opportunities
**Manage learners online interactions** Provide opportunities for learners to share learning outcomes whilst managing incorrect or misleading comments	Anticipate specific groups of learners’ needs where possible during course design (e.g., alternative health backgrounds) Read and respond to learner discussions quickly in forums Judiciously prompt learners when posts go off-topic Provide informal weekly feedback to address common themes arising from discussions Positively reinforce learner comments which are consistent with course messages Provide activities or suggestions to promote active learning for learners who feel uncomfortable posting comments or ideas Ask learners throughout the course how the acquired knowledge will improve health-related behaviors and share their strategies
**Impact on knowledge/behavior** Plan a robust evaluation process ensuring both active and silent learners are included	Be resilient when evaluating learners’ critical comments Monitor discussion forums to gauge learner satisfaction Incorporate short end-of-week confidential evaluations to evaluate “silent” learner perceptions

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
