# Peer review of "Promoting Evidence Based Nutrition Education Across the World in a Competitive Space: Delivering a Massive Open Online Course"

_nutrients, 2020, doi:10.3390/nu12020344_

Round 1
Reviewer 1 Report
Dear authors,
Thank you so much for all your effort putting in this extensive work!
The strength of this research is the large international sample size of participants on the course as well as on the interviews pre and post the course.
You did a very good job in evaluating the large amounts of free answers given by the participants.
The reader gets a clear understanding of which didactic methods were prefered and apricated by the participants.
Nevertheless some information is missing and others needs more explanation.
TITLE
The title implements an all over evaluation of MOOC. But the article gives only insight in the evaluation of the didactic tools used in MOOC. The reader does not learn about understanding the content and implementing the content into daily life situations of the participants. To clarify the content of the article you might change the title.
MATERIALS AND METHODS
Please answer the following questions and add them to your text:
- What was the learning target for the participants of the course? Knowledge and/or increase of health related behavior? Other?
- How much time did it take in average to complete one lecture for a participant? How many lectures did the participants have to work through (in average one for each week or one for each day)?
- Why did you choose exactly these topics and why not other ones (e.g. diet pyramid)?
- How did you measure achievement besides finishing the course? Was there an exam or test at the end? If not, please explain why.
- Was there a certificate for the participants who completed the course?
Line 96: “Supplementary figure 1” was named as “Table S2 Course outline” in my materials.
RESULTS
Please answer the following questions and add them to your text. If you cannot answer them, please explain why.
- Registration of countries: Were there other countries with 2 or more percent? If yes, please list their names.
- What were the reasons for participation?
- What social status did the participants came from?
- What educational level did the participants present?
- What items were disliked? (3.2. Post-course evaluation: Learning Preferences)
DISCUSSION
Please discuss in more detail why increase of knowledge and increase of health related behavior is not measured and evaluated, since these are the most important parameters for evaluation a course.
Author Response
Thank you very much for your constructive feedback and we believe our responses to your questions will provide clarity and detail, thus improving the manuscript. We are most appreciative.
TITLE
The title implements an all over evaluation of MOOC. But the article gives only insight in the evaluation of the didactic tools used in MOOC. The reader does not learn about understanding the content and implementing the content into daily life situations of the participants. To clarify the content of the article you might change the title.
RESPONSE: Thank you for this feedback. We chose to keep the title broad as learners provided their views regarding on many aspects of the course, not only the didactic components. For example, there was a significant portion of the results and discussion related to learners using the online forums (which is not didactic, rather interactive). There were also results related to how learners intended to use the information they learned in the course. To avoid misleading future readers, we have changed the title to Promoting evidence based nutrition education across the world in a competitive space: Delivering a Massive Open Online Course
MATERIALS AND METHODS
Please answer the following questions and add them to your text:
- What was the learning target for the participants of the course? Knowledge and/or increase of health related behavior? Other?
RESPONSE:
The learning targets for participants are both knowledge and behaviour which are described in lines 92-93 “aiming to promote health literacy and eating patterns”. Supplementary Table S2 outlines each topic.
- How much time did it take in average to complete one lecture for a participant? How many lectures did the participants have to work through (in average one for each week or one for each day)?
RESPONSE:
Supplementary Table S2 outlines the lectures and activities for the 3 week course. We apologise for incorrectly labelling this “Supplementary Figure 1” (line 96) in the text which we will amend. It is expected learners spend 4 hours per week which we have now added to the manuscript (line 129).
- Why did you choose exactly these topics and why not other ones (e.g. diet pyramid)?
RESPONSE:
We aimed to make the content applicable to global audiences while addressing topical nutrition issues in the media. We have described this in lines 90-93. National and international dietary guidelines were one of the topics in week 3 (Supplementary Table S2) and more detail summarising course content is in line 128.
- How did you measure achievement besides finishing the course? Was there an exam or test at the end? If not, please explain why.
RESPONSE:
Generally, MOOCs can be viewed as “the new textbook” [Young, Jeffrey R. (27 January, Chronicles of Higher Education, 2013) and as such are not used as traditional achievement-based education. We have now provided more detail regarding connectivst learning which further explains the value of exploring learner engagement. We have elaborated on this further in lines 78-87. Quizzes were interspersed throughout the course so learners could test their own knowledge (added to lines 131-132. The course mentor monitored comments in the discussion forums throughout and provided information where required. Further research is underway which is gathering data about learner self-reported intake as we acknowledge that MOOCs lack evaluation and learning outcome data (lines 372-375).
- Was there a certificate for the participants who completed the course?
RESPONSE:
Certificates of participation were available to purchase to those who completed all modules of the MOOC (line 133-134)
Line 96: “Supplementary figure 1” was named as “Table S2 Course outline” in my materials.
RESPONSE:
Apologies, we have now amended the text
RESULTS
Please answer the following questions and add them to your text. If you cannot answer them, please explain why.
- Registration of countries: Were there other countries with 2 or more percent? If yes, please list their names.
New Zealand was the only country with 2 percent or more, we have added this to the text.
- What were the reasons for participation?
These included wishing to learn about the subject (n=12010, 99%) and the flexible MOOC format suiting their lifestyle. About half (n=6126, 54%) wanted to improve job performance or career prospects. See lines 170-173.
- What social status did the participants came from?
This was not measured for a variety of reasons. These included the difficulties assigning social status to be relevant for 158 countries and questions investigating income, etc were considered to be onerous and invasive. The free aspect of MOOCs being available to everyone in spirit contradicts gathering this sort of information
- What educational level did the participants present?
These data were not collected, however previous nutrition experience was gathered (with 11 % reporting they had studied the subject of nutrition at university) and summarised in lines 168-170. About half had taken an online course previously (line 168)
- What items were disliked? (3.2. Post-course evaluation: Learning Preferences)
In the quantitative survey, disliked/strongly disliked items accounted for <2% of all responses, except for reading comments by other learners (n117, 5%) and discussing things online with other learners (n178, 8%). We have added this to lines 183-184. From the qualitative data, the items both liked and disliked were thematically analysed and presented in the results. Most and least liked responses were synthesised together as learners reported the same items for both.
DISCUSSION
Please discuss in more detail why increase of knowledge and increase of health related behavior is not measured and evaluated, since these are the most important parameters for evaluation a course.
The aim of this research was to to (1) provide an overview of a nutrition MOOC and its’ content; (2) describe course participation rates and participant backgrounds; (3) report on learners preferred learning modes and content, and (4) share our learnings and recommendations. We agree that knowledge and behavior are important parameters and this is a limitation of current MOOC research as described in lines 372-375. In the online space, with thousands of learners, the research team believed learner engagement and satisfaction was the initial priority to measure given the connectivist learning approach adopted. If online learners are not engaged, regardless of learning outcomes and behavior, they will quickly move to other platforms that are more enticing. We hope that by providing more detail regarding connectivist MOOC learning in the introduction, the lack of knowledge/behavior measurements now make more sense.
The research team are currently investigating the very question raised, but it is not in the scope of this particular research to measure behavior or knowledge.
Reviewer 2 Report
This paper is overall well done and well organized. The largest contribution to MOOC research is Table 1 with thematic results and suggested best teaching practices, founded in the authors' empirical results. The introduction could be a bit deeper in MOOC literature, especially in lines 67-83. Include more literature about the design of MOOCs, that well-educated adult learners typically enroll in MOOCs, types of topics they typically cover, examples of the different platforms, etc. Explain c-MOOC/connectivism in more detail. Line 79 says "This MOOC" - which MOOC does this mean? Possibly refer to the tile of the MOOC studied. Methods section - give more details about content that was in each module - what was the topic focus of each module? Line 94 - explain connectivism principles and First Principles in more detail - how did the authors conceptutally combine these two different designs? Discussion - reinforce findings from MOOC literature about authors' results that learners have a variety of motivations for participating, as well as prefer the video content.
Author Response
This paper is overall well done and well organized. The largest contribution to MOOC research is Table 1 with thematic results and suggested best teaching practices, founded in the authors' empirical results.
Thank you, and thanks for the time you have take to provide constructive feedback and suggestions which we have responded to (below).
The introduction could be a bit deeper in MOOC literature, especially in lines 67-83.
Include more literature about the design of MOOCs, that well-educated adult learners typically enroll in MOOCs, types of topics they typically cover, examples of the different platforms, etc. Explain c-MOOC/connectivism in more detail.
RESPONSE:
We have added more detail regarding MOOC design, connectivism, typical learners, and the breadth of MOOCs available to provide more context (lines 73-95)
Line 79 says "This MOOC" - which MOOC does this mean? Possibly refer to the tile of the MOOC studied.
RESPONSE:
We have amended this to describe the MOOC in this study and have included the MOOC title (line 98).
Methods section - give more details about content that was in each module - what was the topic focus of each module?
RESPONSE: This was outlined in Supplementary Table S2 (apologies it was incorrectly labelled Figure S1 in the text). We have provided a summary of the week by week content in the text also (lines 124-129).
Line 94 - explain connectivism principles and First Principles in more detail - how did the authors conceptutally combine these two different designs?
We have added this explanation to lines 114-123.
Discussion - reinforce findings from MOOC literature about authors' results that learners have a variety of motivations for participating, as well as prefer the video content.
Thank you for this suggestion. We have included further literature to reinforce our findings, including motivations for participating and connecting, as well as video content.